# Glycosphingolipids in Diabetes, Oxidative Stress, and Cardiovascular Disease: Prevention in Experimental Animal Models

**DOI:** 10.3390/ijms232315442

**Published:** 2022-12-06

**Authors:** Amrita Balram, Spriha Thapa, Subroto Chatterjee

**Affiliations:** Department of Pediatrics, Cardiology Division, Johns Hopkins University School of Medicine, Baltimore, MD 21287-3654, USA

**Keywords:** diabetes, sphingolipids, insulin, lipidomics, inflammation

## Abstract

Diabetes contributes to about 30% morbidity and mortality world-wide and has tidal wave increases in several countries in Asia. Diabetes is a multi-factorial disease compounded by inflammation, dyslipidemia, atherosclerosis, and is sometimes accompanied with gains in body weight. Sphingolipid pathways that interplay in the enhancement of the pathology of this disease may be potential therapeutic targets. Thus, the application of advanced sphingolipidomics may help predict the progression of this disease and therapeutic outcomes in man. Pre-clinical studies using various experimental animal models of diabetes provide valuable information on the role of sphingolipid signaling networks in diabetes and the efficacy of drugs to determine the translatability of innovative discoveries to man. In this review, we discuss three major concepts regarding sphingolipids and diabetes. First, we discuss a possible involvement of a monosialodihexosylceramide (GM3) in insulin–insulin receptor interactions. Second, a potential role for ceramide (Cer) and lactosylceramide (LacCer) in apoptosis and mitochondrial dysfunction is proposed. Third, a larger role of LacCer in antioxidant status and inflammation is discussed. We also discuss how inhibitors of glycosphingolipid synthesis can ameliorate diabetes in experimental animal models.

## 1. Introduction

Diabetes mellitus (DM) is a metabolic syndrome that affects our body’s use of glucose. In the US, about 37.3 million people are diagnosed with diabetes. Globally, about 700 million people are predicted to have diabetes by 2045 [1]. DM is a chronic disease that is characterized by elevated blood glucose levels, also known as hyperglycemia. Hyperglycemia can be induced by insulin resistance and develop into pre-diabetes and type-II diabetes. Pro-longed hyperglycemia can have devastating effects on the body’s organ systems and contribute to cardiovascular disease co-morbidities.

Diabetes is split into four main types: type-I diabetes (T1D), type-II diabetes (T2D), types of diabetes due to other causes, and gestational diabetes [2,3]. TD1 is commonly known as “insulin-dependent diabetes” or “juvenile-onset diabetes” and is caused by the autoimmune destruction of pancreatic *β*-cells. T2D, or “insulin-dependent diabetes” or “adult-onset diabetes”, is the most prevalent (90–95% of all diabetes) and is caused by insulin resistance [2]. Other specific types of diabetes include monogenic diabetes syndromes, diabetes caused by diseases of the exocrine pancreas, and drug- or chemical-induced diabetes [2]. Gestational diabetes is diagnosed during pregnancy and poses a risk to the mother and fetus [2]. 

There are several stages in diabetes. The initiation phase, lending to patients becoming pre-diabetic, is characterized by a rise in blood glucose levels from below 100 mg/dL (normal adult male) to 110 mg/dL (pre-diabetic). The progressive phase, the transition of pre-diabetic to diabetic, involves not only a rise of blood glucose levels, but also certain sphingolipids (see below) that exacerbate oxidative stress and inflammation. The next phase involves massive changes in lipid and lipoproteins in the blood e.g., increased levels of cholesterol, low-density lipoproteins (LDL) cholesterol, and levels of triglycerides, as well as decreased levels of high-density lipoprotein (HDL) cholesterol, collectively called “metabolic syndrome”. Obesity and atherosclerosis set in, contributing to a multi-organ disease e.g., kidney, heart lungs, and the circulatory system. 

Pancreatic *β*-cells are the producer of insulin, thus destruction of these cells e.g., via apoptosis and/or oxidative stress, may contribute to diabetes. In the insulin signaling pathway, the heterodimer insulin receptor (IR) undergoes autophosphorylation of three tyrosine residues after the alpha subunit activates the beta subunit. This results in the phosphorylation of insulin receptor substrate (IRS) proteins and the downstream activation of the Akt/phosphoinositide 3-kinases (P13-kinase)/mTOR pathway and their associated phenotypes [2]. While the molecular mechanism of insulin resistance is still being understood, studies have shown that protein kinase B (PKB) or Akt2 knockdown has resulted in a T2D phenotype [4,5]. Additionally, mutations or hyper-serine phosphorylation of the insulin receptor substrate-1 (IRS-1) protein and knockdown of the IRS-1 gene have resulted in a state of insulin resistance in human, mouse, and rodent models [6,7,8,9,10,11]. Insulin resistance primarily occurs in the muscles, adipose, and liver tissue. Insulin resistance is a state of insensitivity of cells to insulin and is commonly caused by defects in the insulin signaling pathway. Risk factors for diabetes, such as obesity, diet, genetics, and smoking, can all contribute to insulin resistance. Obesity leads to increased lipid metabolite and fatty acid accumulation, which can contribute to insulin resistance in the liver and skeletal muscle [12]. Insulin resistance results in increased *β*-cell insulin production, which leads to weight gain, perpetuating a cycle of insulin resistance [13,14]. Other causes of insulin resistance are abnormal glucose transporter type 4 (GLUT4) function or translocation and mitochondrial dysfunction [15].

Sphingolipids (SLs) are a class of lipids which contain a backbone of sphingoid bases and a polar sphingosine head. The two main classes of sphingolipids include phosphosphingolipids (PSLs) and glycosphingolipids (GSLs). This review article will focus on the role of GSLs. GSLs (Figure 1) have been implicated in various disease pathologies, including diabetes [16]. GSLs function as second messengers in several pathways leading to major phenotypes i.e., proliferation, adhesion migration, autophagy, apoptosis, and mitochondrial function [17]. Consequently, inhibition of glycosphingolipid synthesis has shown promise in mitigating these phenotypes in animal models of disease [16]. 

The literature is too vast to present numerous studies using cultured cells and multiple animal models of diseases dealing with T2D, as well as SL-relevant signaling pathways. Hence, the readers are referred to excellent review articles by pioneers in the SL field (Hannun, Spigel, and our IJMS review article) [15,16,18].

In this review, we shall focus on SLs in the initiation and progression of T2D, as well as SL-centric drugs, which have shown promise to mitigate T2D and metabolic syndrome in animal models by managing blood glucose levels and obesity.

## 2. Lipoproteins and Sphingolipids 

We and others have shown that most, if not all, SLs are carried in blood associated with lipoproteins, and very little, if any, are found to be associated with lipoprotein–deficient serum [18].

Among lipoproteins, very-low-density lipoprotein (VLDL) carries the bulk of triglycerides and cholesterol. LDL and HDL carry the bulk of cholesterol and SLs, including GSL and SM. However, HDL is particularly enriched in sphingosine-1-phosphate. Thus, LDL-receptor deficiency, as in patients with familial hypercholesterolemia, also raises the serum level of LDL and lipids associated with it. Additionally, the shedding of GSL e.g., LacCer in urinary sediments, is markedly increased in these patients [19].

GSLs for LacCer associated with LDL are delivered to normal peripheral cells via receptor–dependent endocytosis. In contrast, they are delivered via receptor-independent/ scavenger receptor pathway in LDL-receptor-deficient cells [20]. The exchange of SLs from lipoproteins to cells has also been shown [21]. Herein, SLs impart a diverse set of signaling pathways depending on the cell type and the end point affects phenotypes, e.g., proliferation, migration, angiogenesis, apoptosis, and most importantly inflammation.

## 3. Sphingolipid Pathways in Diabetes

Sphingolipids are generated through two main pathways: de novo synthesis or salvage [22]. In the de novo synthesis pathway, the condensation of palmitoyl-coenzyme A (palmityl-CoA) and L-serine by serine palmitoyltransferase (SPT) forms 3-ketosphinganine, which gets reduced to sphinganine by 3-ketophinganine reductase [23]. The addition of a fatty acid coverts sphinganine to ceramide. In the salvage pathway, sphingomyelinases (SMase) and glucosylceramidase (GCase) are involved in the creation of ceramide [22]. Ceramide can lead to the formation of several glycosphingolipids via the sequential addition of glucose or galactose sugars by glycosyltransferases. Additionally, ceramide is an intermediate for sphingomyelin (SM). The addition of glucose to ceramide results in glucosylceramide, which can form LacCer via the transfer of galactose from uridine-diphosphate galactose (UDP-galactose). LacCer plays a pivotal role in the biosynthesis of complex glycosphingolipids as it serves as a precursor to, e.g., GM3 gangliosides, sulfatides, and globostriosylceramide [24] (Figure 1). Sphingolipids can also be generated by the catabolism of sphingomyelin, a major membrane-bound phospholipid, via the action of acid sphingomyelinases and neutral sphingomyelinases. The product ceramide is then glycosylated to form GSLs or catabolized to produce sphingosine and fatty acids (Figure 1). The catabolism of complex GSL via the action of respective glycosidases can also generate simpler GSLs.


**Diabetes and Ceramide**


Ceramides, comprised of sphingosine and a fatty acid, represent the non-polar component of GSLs and are found in the cell membrane bilayer [24]. They are highly enriched in the mammalian skin and provide hydration homeostasis [25]. Ceramide is also known to induce cell apoptosis via caspase activation [16,17]. In diabetes, ceramide contributes to apoptosis, inflammation, mitochondrial dysfunction, and an increased state of insulin resistance. Increased ceramide levels have been observed in various tissues in patients with obesity and T2D in animal models [26]. Additionally, muscle biopsies from insulin-resistant patients had increased levels of ceramide species [27]. Ceramide disrupts the insulin signaling pathway in three main ways (Figure 2). 

First, ceramide generation is driven by the upstream activation of the pro-inflammatory toll-like receptor 4 (TLR4) (Figure 2) [28]. A study that used a TLR4 mutant line of hematopoietic cells to eliminate TLR4 function in mice found increased insulin sensitivity and reduced inflammation [29]. Therefore, TLR4 is not only a key receptor in the inflammatory pathway but is also implicated in the development of insulin resistance [29]. The mechanism through which TLR4 leads to a state of insulin resistance is through increased ceramide synthesis via IKKβ and NF-κB. Ceramide then inhibits the Akt/PKB pathway leading to reduced glucose uptake, lipolysis, gluconeogenesis, and anti-glycogen synthesis, as well as phenotypes of insulin resistance (Figure 2) [30,31]. 

Second, ceramide leads to insulin resistance via the activation of PP2A and protein kinase C (PKC) isoform PKCζ, which inhibit the Akt/PKB pathway. In skeletal muscle, ceramide activates PKC which combines with PKB to inhibit the signal transduction pathway [26]. Normally, insulin deregulates PKB–PKCζ interactions to activate PKB alone [32]. Ceramide, however, inhibits their dissociation and the activation of PKB [32]. Additionally, ceramide increases Akt’s association with PP2A, which leads to Akt dephosphorylation and deactivation [33]. 

Lastly, in diabetes, ceramide induces *β*-cell apoptosis in the pancreas [34]. Ceramide was found to increase the release of cytochrome-C, which triggers apoptotic pathways in pancreatic *β*-cells [34]. Ceramide was also associated with increased mitochondrial damage, oxidative stress, and inhibition of ion channels, which were all associated with apoptosis in *β*-cells [34]. Pancreatic *β*-cells produce insulin and therefore play a major role in diabetic pathophysiology. 

In a diabetic rat model, ceramide’s pre-cursor sphingolipid, palmitate, has been correlated with an increase in *β*-cell apoptosis via the activation of ceramide [35]. Ceramide is also implicated in the P13K pathway. Activation of ceramide via palmitate leads to ceramide inhibiting Akt/PKB via PPA2 and PKC (Figure 2) [33]. Activation of PPA2 by ceramide dephosphorylates Akt, thus inhibiting its translocation to the plasma membrane. Inhibition of Akt/PKB blocks the insulin receptor pathway and insulin action, as described above. Additionally, a study found that ceramide downregulates the gene expression of GLUT4 by about 60% in adipocytes treated with ceramide [36]. Therefore, ceramide can affect glucose production at the transcriptional level as well. 

Studies targeting enzymes in the ceramide synthesis pathway, e.g., ceramide synthase, have been shown to decrease ceramide levels and therefore better regulate glucose homeostasis in Zucker diabetic fatty rats [37].

In summary, reducing ceramide levels in the liver in T2D may be helpful in the management of diabetes. However, recent studies show that reducing LacCer levels may be sufficient to reduce blood glucose, cholesterol, and triglyceride-rich lipoproteins and reduce body weight in a mouse model of T2D (*db*/*db*) mice without reducing ceramide levels [16], see below.
Figure 2Insulin receptor-dependent and -independent pathways by which GM3 and ceramide, respectively, may be implicated in diabetes. IR binding to IRS triggers a signaling cascade that activates P13K. P13K signaling leads to downstream events such as GLUT4 translocation and activation of the Akt/PKB pathway. These pathways lead to cell proliferation, protein synthesis, lipid synthesis, gluconeogenesis, and activation of glycogen synthesis pathways. Ceramide and GM3 sphingolipids interrupt the insulin signaling pathway, creating a state of insulin resistance. Reprinted/adapted with permission from Ref. [38] 2022, Dr. Essam [38].
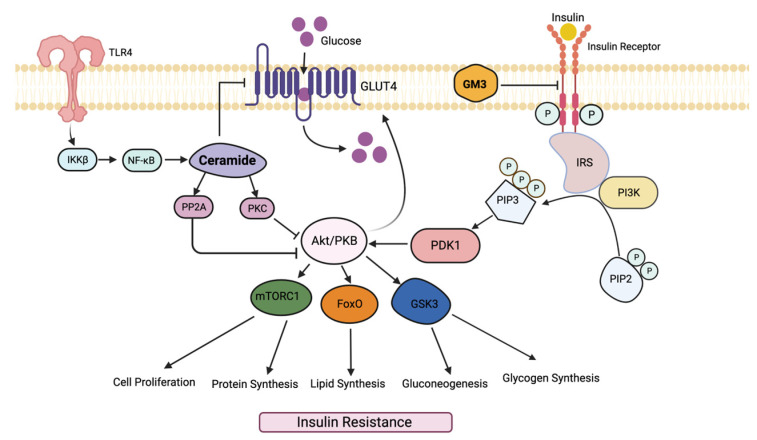




**Diabetes and GM3 Ganglioside**


GM3 is localized in the cell plasma membrane and is formed by GM3 synthase from lactoslyceramide (Figure 1) [23]. GM3 plays a major role in the insulin resistance pathway by blocking insulin receptor function [39]. In Zucker a *fa*/*fa* rats and *ob*/*ob* mouse insulin resistant animal models, high GM3 synthase mRNA levels were found, implicating GM3 in the insulin resistance pathway [39]. One mechanism by which GM3 induces insulin resistance is via the activation of an inflammatory pathway induced by tumor necrosis factor α (TNF-α). In adipocytes, TNF-α downregulates the expression of vital parts of the insulin signal transduction pathway, such as insulin receptor substrate 1 and GLUT4 [40]. It was also found that GM3 can suppress glucose uptake on its own, therefore mimicking the role of TNF-α [40]. Most importantly, treatment of adipocytes with 0.1 nM TNF-α upregulated GM3 synthesis. Treatment of 3T3-L1 adipocytes with GM3 repressed phosphorylation of the insulin receptor, which led to the repression of the insulin receptor pathway [39]. Additionally, a glucose and insulin tolerance test determined higher insulin sensitivity due to greater insulin receptor phosphorylation in a GM3 synthase knockout mouse model [41]. Several studies have also demonstrated that increased GM3 levels in the adipocyte membranes leads to IR and IRS-1 uncoupling due to IR dissociating from the membrane microdomain caveolin-1 (Cav-1) [42,43]. GM3 inhibits insulin receptor signaling to IRS-1, therefore creating a state of insulin resistance by preventing downstream signaling events (Figure 2). 


**Diabetes and Sphingosine-1-Phosphate**


Spingosine-1-phosphate (S1P) is another crucial sphingolipid that plays a role in cell migration, proliferation, and apoptosis [23]. S1P is formed from the de-acylation of ceramidases and then the phosphorylation of sphingosine by the sphingosine kinases 1 and 2 (SphK1 and SphK2) [44]. Upon formation, S1P interacts with five different GPCR receptors (S1PR1-5) to activate downstream signaling cascades, such as insulin signaling pathways [45]. 

In addition to S1P’s close relationship with ceramide, S1P itself has been associated with obesity and T2D onset. In obesity animal models fed a high fat diet (HFD), studies have noted increases in S1P levels in the liver, skeletal muscle, and plasma via increased SphK1 and SphK2 expression [18]. 

S1P’s role in diabetes is controversial and requires further research. Previously, it has been shown that SIP can stimulate and suppress diabetic pathophysiology. S1P acts as a mediator to protect against insulin resistance by activating the Akt pathway and increasing mitochondrial function when bound to apoM [44]. Additionally, S1P can activate AMPK and increase the survival of pancreatic beta cells, increase nutrient uptake, nutrient utilization, and mitochondrial proliferation [46]. 

The SIP1/SphK1 axis is known to stimulate diabetic pathophysiology in adipose tissue. In adipocytes, the PI3K/Akt signaling pathway plays a major role in regulating glucose uptake. S1P and SphK1 were found to inhibit the cAMP/PKA signaling pathway, creating a state of insulin resistance in adipose tissue. The SphK1/S1P axis also inhibited chronic inflammation and the release of chemokines/cytokines, which play a vital role in insulin-resistant pathogenesis [46]. A study with animal models of type I diabetic rats and mice demonstrated increased S1P blood levels [47]. When the mice were given a S1P receptor antagonist medication it led to reduced blood glucose levels, *β*-cell apoptosis, and an increased insulin-to-glucose ratio [46]. Therefore, S1P and its receptors are possible targets in diabetes. 

Additionally, S1P plays a role in pancreatic *β*-cell insulin regulation. A study demonstrated that the knockdown of SphK1 reduced insulin secretion in INS-1832 cells [48]. SphK2 knockdown resulted in no insulin release at all in MIN6 cells, which would result in hyperglycemia [48]. However, in *β*-cells, SphK1 and SphK2 have antagonist effects. SphK2 induces apoptotic effects and *β*-cell lipotoxicity. The receptor S1PR2 inhibits the PI3K/Akt pathway, which prevents *β*-cell regeneration, cell survival, and growth. On the other hand, SphK1 leads to enhanced *β*-cell survival and growth. The S1PR receptor, when activated, leads to insulin secretion and activation of the PI3K/Akt pathway [45].

In summary, S1P levels are governed by SphK1, SphK2, and S1P receptors. The duality of these enzymes and receptors’ effects on the insulin signaling pathway makes S1P’s role in diabetes controversial.


**Diabetes and Lactosylceramide**


In diabetes, the glycosylation of ceramide to form LacCer leads to mitochondrial dysfunction, causing lower energy production, reduced respiration, and Ca^2+^ retention [16]. MAMs, or mitochondria-associated endoplasmic reticulum, connect the ER to the mitochondria and are where lipid synthesis, mitochondrial dysfunction, and apoptosis can occur [49]. A study with streptozotocin-induced type I diabetic mice found an accumulation of LacCer in the ER and the mitochondria [50]. In the same mice model, adding 100 μM LacCer to mitochondria led to reduced mitochondrial respiration and calcium retention capacity (CRC) [49]. Therefore, increased LacCer production worsens diabetes through inhibiting a cardiac cell’s ability to retain calcium and process energy due to reduced respiration and increased obesity, such as in *db*/*db* mice fed regular mice chow (see below).

While ceramides and SM were associated with disease pathology, hexosylceramide and LacCer were suggested to have an immunomodulatory role in disease pathology [51,52]. The cited Chew paper above and Huynh paper support this [53,54].


**Diabetes and Sphingomyelin**


Sphingomyelin is the major SL in mammalian tissue and blood and constitutes the major lipid species in brain [55]. Sphingomyelin synthesis is carried out by a family of SM synthases (SMS) via the transfer of a phosphoryl–choline group to ceramide, e.g., SMS1 and SMS2 [56]. A third SMS called SMS-related protein catalyzes the synthesis of ceramide phosphoethanolamine. 

Nearly 70% of the cellular SM is localized within the plasma membrane wherein it decorates the lipid rafts along with other SLs, e.g., LacCer and a ganglioside, GM3, as well as the insulin receptor. Thus, perturbation of the plasma membrane, and/or alterations of the levels of one or more of these SL may well affect diabetes. 

The role of SMS2 and serine–palmotyltranseferase-2 in insulin metabolism arrived from studies using mice deficient in these two enzymes [57]. The level of SM in the plasma membrane in SMS2-deficient mice was decreased but the ceramide level was increased. A critical physiologically important observation made in this study was that these mice had increased sensitivity to insulin and increased resistance to obesity due to a high fat diet. However, the level of glucosylceramide and GM3 remained unchanged, and these observations were replicated with the SPtlc2-deficient mice as well. 

Chew et al. used large scale lipidomics to identify the hexosylceramides d 16:1 and d 18:1 to significantly associate with obesity and T2DM [54]. They suggested that these SLs may account for the pathology of obesity. They also reported that SM d: 16:1/18:0 and SM d18:1/18:0 significantly correlated with the occurrence of diabetes irrespective of sex, age, and body mass index (BMI). As the plasma samples were derived from Asian American patients living in Singapore, and as they are known to have higher levels of SMs compared to Americans of European ancestry, this may well contribute to conclusions drawn due to susceptibility and early onset of disease, environmental differences, and genetic differences in the Chew et al. study compared to other studies [58].

## 4. Sphingolipid Inhibitors Mitigate the Pathology of Insulin Resistance

As described above, sphingolipids play a vital role in the pathology of insulin resistance and therefore are potential targets for its mitigation. One study observed insulin resistance with H9C2 cardiomyocytes and adult mice treated with insulin and myriocin, a serine–palmitoyl transferase inhibitor (Figure 1) [59]. They found that insulin affected heart mitochondrial respiration in the cardiomyocytes likely due to increased mitochondrial fission [59]. Furthermore, they found that insulin increased ceramide levels in a time-dependent manner, implicating it as a mediator of cardiac deficiencies [59]. Myriocin alone can serve as an immunosuppressant and decrease fatty acid metabolism and thus reduce liver inflammation and steatosis by regulating genes, e.g., PPAR-α, and Fabp1.

Glycosphingolipid synthesis inhibitors such as D-thero-1-phenyl-2-decanoylamino-3-morpholino-1-propanol (D-PDMP) and D-thero-1-(3,4,-ethylenedioxy) phenyl-2-palmitoylamino-1-pyrrolidino-1-propanol (D-EtDO-P4) have shown to cause a decline in GM3 levels and an increase in Akt1 kinase phosphorylation [60]. Inhibiting GM3 synthesis via D-PDMP and D-EtDO-P4 increased insulin-resistant autophosphorylation in addition to Akt1 phosphorylation by 286% in HepG2 cells [60]. Previously, D-PDMP was known to specifically inhibit the activity of GlcCer synthase (Figure 1). Treatment of adipocytes with D-PDMP, which inhibits glucosylceramide synthase, reversed TNF-α’s effect on suppressing IRS-1 phosphorylation [39]. However, we and others showed that D-PDMP not only inhibited the activity in purified human kidney LacCer synthase and reduced LacCer generation, but also in various organs derived from multiple animal models of cardiovascular disease and T2D [16,61]. The pathophysiological role of GM3 involving toll like receptor-4 binding in adipose tissue in metabolic syndrome has been elegantly shown recently [62].

## 5. Inhibition of LacCer Synthesis Can Mitigate the Pathology of Type II Diabetes by Reducing Blood Glucose, Body Weight, and Inflammation

Another important aspect of type II diabetic and cardiovascular disease in man is the observation of a “trifecta” consisting of a. increased blood levels of LDL cholesterol, b. increased triglyceride levels, and c. decreased HDL cholesterol level. This phenotype, collectively known as “metabolic syndrome”, may play an important role in type II diabetic patients’ disease progression and cardiovascular health. Since nearly all the sphingolipids are carried on lipoproteins, an increase in the level of cholesterol-rich and triglyceride-rich lipoproteins also raised the level of these sphingolipids, which adversely affects the progression of T2D and heart disease (Figure 2 and Figure 3). While an increase in the level of GM3 and other gangliosides may hamper insulin receptor signaling, ceramide could bring about mitochondrial dysfunction, reduced energy production, reduced respiration and increased apoptotic death. On the other hand, increased levels of LacCer would activate superoxide generation and multiple signaling pathways leading to inflammation, proliferation, adhesion, migration, angiogenesis apoptosis and increased oxidative stress [16].

LacCer initiates the first step in the “inflammatory pathway” by two independent mechanisms; the first is by recruiting neutrophils and monocytes to the inflamed tissue, e.g., diabetic wounds, liver, and gastrointestinal tract. The infiltration of neutrophils and monocytes to an inflamed tissue is initiated by a simultaneous increase in the expression of cell adhesion molecules due to the release of cytokines. For example, in endothelial cells, LacCer exerted a dose-dependent increase in the expression of intercellular cell adhesion molecule (ICAM-1) and platelet cell adhesion molecule (PECAM-1), and in freshly obtained neutrophils/monocytes from human subjects, LacCer increased the expression of CD-11b. ICAM-1 serves as a receptor to its ligand CD11b and thus initiates the adhesion of circuiting neutrophils and monocytes to the endothelium, followed by their intravasation and entry into the sub-endothelial space. Herein, neutrophils can release the load of LacCer they carry (Figure 3). In turn, LacCer can produce superoxides and convert LDLs trapped in the matrix to oxidized LDLs (ox-LDL). Such ox-LDL is taken up by monocytes via a scavenger receptor-dependent pathway, e.g., via CD36, SRA-1 and Lox-1 to form foam cells, followed by complications that adversely affect cardiovascular functions [16]. In the second mechanism, the endogenously derived LacCer can activate neutrophil/monocyte cytosolic phospholipase-C via phosphorylation via the PKC/MEK/ERK pathway to degrade a major cell membrane phospholipid phosphatidylcholine to release arachidonic acid and lysophophatidylcholine, a potent lysogenic agent that causes apoptosis (Figure 3).

On the other hand, arachidonic acid serves as a precursor to a family of compounds called eicosanoids, which includes prostaglandins. In particular, prostaglandin E2 plays a central role in inflammation. Interestingly, these phenotypes in type II diabetic patients can be reproduced in a mouse model of type II diabetes, *db*/*db*, fed regular mouse chow. 

Thus, the availability of *db*/*db* mice and the observations above collectively rationalized investigators to examine the effects of lowering the load of GSL by feeding a GSL synthesis inhibitor on body weight, dyslipidemia, and most importantly blood glucose levels. In this study, a biopolymer-encapsulated D-PDMP (BPD) was fed by oral gavage to 30 week-old *db*/*db* mice daily for 6 weeks. The advantages of BPD over D-PDMP were the following: a. BPD is at least 10 times more efficacious in lowering the levels of GSL and therefore is cost-effective compared to D-PDMP; b. BPD was well-tolerated when fed at 100 times the optimal dose for long periods of time (6 months); c. it increased t1/2 time from <1 h (D-PDMP) to 48 h (BPD) [63]. 

Following treatment with BPD in *db*/*db* mice it was observed that the body weight, LDL cholesterol, and triglyceride levels were all decreased. In contrast, the levels of HDL cholesterol increased. Most importantly, treatment significantly reduced blood glucose levels. As expected, treatment markedly reduced the level of GlcCer and LacCer, but not ceramide. Thus, the latter observation suggests that in *db*/*db* mice, T2D can be managed independently of lowering ceramide levels.

Molecular studies further substantiated a mechanistic explanation accompanying the reversal of dyslipidemia in *db*/*db* mice treated with BPD. For example, in the liver, treatment upregulated the genes implicated in regulating cholesterol homeostasis, thus, the expression of *HMG-CoA reductase*, *Srebp2*, and *ApoA-1*, and the genes implicated in cholesterol, efflux, e.g., *CD36* and *SRB1,* were increased. As cholesterol is a precursor to bile acids, *Cyp7*, a gene encoding the rate-limiting enzyme in bile acid synthesis from cholesterol, was examined. Treatment raised the expression of *Cyp7*. Similarly, the expression of genes which facilitate bile acid secretion, e.g., *Abca1*, *Lxr*, and *Fxr,* were also increased. Thus, treatment with BPD accelerated the conversion of cholesterol to bile acids as well as its secretion. 

Recent studies have shown that lysosomal cholesterol activates mTORC-1 via an SLC38A-9 Niemann-Pick C1 (NPC-1) signaling complex, contributing to cell proliferation and cell growth (Figure 3) [64]. Western immunoblot assays and qRT-PCR assays revealed that in *db*/*db* mice treatment with BPD increased lysosomal cholesterol degradation and consequently decreased the protein and gene expression for NPC-1 and mTORC-1 and thus helped reduce cell growth and reduce body weight (Figure 3).

Decreased levels of triglycerides in *db*/*db* mice treated with BPD could have been due to the increased expression of genes involved in triglyceride-rich lipoproteins, including VLDL, and triglyceride metabolism, e.g., VLDLr and lipoprotein lipase. Additionally, there was an increased expression of PPAR-γ, known to regulate fatty acid metabolism.

An important aspect of this study was to examine the effects of lowering GSL levels on genes involved in antioxidant defense. Previous studies implicate SOD1, SOD2, catalase and hypoxia-inducible factor in oxidative stress. Treatment with BPD increased the expression of SOD2 and catalase. Now, SOD2 is of considerable interest as it is a component of the mitochondrial membrane, and LacCer reduces the activity of this enzyme. Thus, as BPD decreased LacCer levels, this may have contributed to increased expression of SOD2 and thus improved antioxidant defense.

Regarding cardiovascular health, echocardiography revealed aortic intima–media thickening and histochemistry revealed ballooning fatty adipocytes and fibrosis in *db*/*db* mice but not in normal C57BL/6J mice aortae. Treatment with BPD reduced the size of lipid-laden adipocytes within the intima and a significant reduction in intima–media thickening. 

In summary, type II diabetes is a complex disease involving multiple genes and phenotypes. Thus, designing drugs for this disease must consider these diverse phenotypes simultaneously in play. At present certain GlcCer synthase inhibitors, e.g., Miglustast and Genz-11263 (1,5-butylamino,1,5 dideoxy-d-glucitol) are prescribed to reduce GlcCer levels in Gaucher’s patients. However, miglustat is contraindicated for use in patients with cardiovascular disease. On the other hand, metformin has been largely successful in lowering blood glucose levels in diabetic patients [65,66,67]. As heart failure and kidney disease is common among type II diabetic patients, treatment with metformin has shown beneficial effects on heart failure. However, metformin is contraindicated in some type I and type II diabetic patients also having chronic kidney disease and/or lactic acidosis [68].

All things considered, pre-clinical studies using BPD in type II diabetic mice reveal that this compound may well meet nearly all the requirements needed for therapeutic use in this disease, as BPD could lower blood glucose, improve antioxidant status, and reduce dyslipidemia, inflammation, and body weight.

## 6. Sphingolipidomics Helps Diagnose and Raise the Predictive Value in Diabetes

Over the past few decades, advancements in mass-spectrometric technologies have accelerated the pace of research into determining novel bio-molecular markers of disease using metabolomics/lipidomics, and this has become a powerful tool to explain the pathophysiological basis of disease, as well as in diagnosis. Since lipids play a central role in obesity, metabolic syndrome, diabetes and atherosclerosis, several studies have employed lipidomics to ascertain biomarkers of these diseases. Moreover, correlations between dyslipidemia and arterial stiffness measured as brachial ankle-pulse wave velocity (PWV) have been drawn as well. For example, a comparative study with obese middle-aged men and normal weight controls was conducted over a period of three years. Metabolomics and lipidomic studies in these individuals revealed that 1. being overweight adversely affects arterial stiffness and thus cardiac function and 2. serum levels of LacCer, L-octanoylcarnitine, systemic blood pressure, and body mass index were found to be independent predictors of arterial stiffness/cardiac function [69]. Earlier, in a mouse model of atherosclerosis, *ApoE*^−/−^ mice fed a Western diet (high fat and high cholesterol diet) increased blood levels of LacCer, which was correlated with increased arterial stiffness, and aortic media–intima media thickening (AOIMT). Inhibiting GSL synthesis not only lowered the level of LacCer and GlcCer but also improved arterial stiffness and AOIMT [70]. 

In another study, plasma lipidomic profiles were assessed to predict cardiovascular outcomes in T2D patients as compared to traditional risk factors in this disease. Herein, 310 plasma lipid species were measured by mass spectrometry in 3779 type II diabetic patients over a period of five years. Four species of lipids, namely sphingolipids, certain phospholipids, cholesterol esters, and GSLs, were associated with cardiovascular death and cardiovascular events. Overall, the inclusion of these four lipid species with the traditional risk factors raised the predictive value of cardiovascular events with statistical significance [70].

Studies show that obesity, insulin resistance, and type II diabetes are associated with a disease called non-alcoholic fatty liver disease (NAFLD). When NAFLD sets roots, it worsens the prognosis of T2D patients. Previously, several animal models of NAFLD have shown the involvement of various sphingolipid species, sometimes dissociating liver steatosis with insulin resistance [71,72,73,74,75], and in other studies, LacCer has been associated with heart defects in diabetic mice [15]. We have correlated increased levels of LacCer in liver of type II diabetic mice with systemic inflammation involving multiple pro-inflammatory molecules, e.g., IL-6, hs-CRP, TNF-α [17]. Sphingolipids have also been analyzed in 21 insulin-resistant obese patients each with and without NAFL. Herein, increased serum and liver total ceramides strongly correlated with TBARS—a measure of oxidation. Moreover, the serum and liver levels of LacCer were markedly increased in NAFL+ patients and were correlated with an increase in the level of the pro-inflammatory cytokine TNF-α, but not IL-6 [76].

Mechanistic studies using mitochondria from diabetic mice livers have suggested that although the ceramide level is increased due to catabolism by a neutral sphingomyelinase (Figure 1), it is not implicated in mitochondrial dysfunction. Rather, when ceramide was glycosylated to LacCer, it produced free oxygen radicals and disrupted mitochondrial function [77].

In this study a glucosyltransferase inhibitor (1R2R)-nonanoic acid [2-(2′-3′-dihydro-benzo [1,4] dioxin-6′-yo)-2hydroxy-1-pyrrolidin-1-ylmethyl-ethyl]-amide -L-tartaric acid salt (Genz-1233346) was found to improve glucose tolerance as well as lower A1C and glucose levels. Herein Genz 1233346 (85 mg/kg) was delivered by oral gavage daily for six weeks to Zucker diabetic fatty rats, known to lose pancreatic *β*-cell function. Treatment with Genz-123336 decreased the level of glucosylceramide and GM3, as well as prevented pancreatic *β*-cell function in this experimental animal model. Mechanistic studies revealed that in these mice, as well as in another mouse model, diet-induced obese mice treatment, improved the phosphorylation of insulin receptors and their downstream signaling functions. However, treatment did not significantly decrease the level of triglycerides, cholesterol, or HDL. Nevertheless, treatment did improve insulin receptor phosphorylation as well as AKT-1 phosphorylation in the muscles of Zucker diabetic rats. Overall, these studies suggest that lowering the levels of GSL improves insulin receptor function [78].

## 7. Conclusions

Although dysfunctional glucose homeostasis is central in T2D, disease progression from a pre-diabetic stage to fulminant diabetes in man may well involve SLs, e.g., LacCer. Accompanying this is inflammation due to a rise in blood levels of pro-inflammatory cytokines and reactive oxygen species, and the infiltration of neutrophils and monocytes into the vascular wall, thus, ushering a decline in cardiovascular functions and aortic media–intima thickening. Metabolic syndrome, defined as a hyperlipidemic state plus cardiovascular complications, involves marked increase in blood levels of lipoproteins as well as lipids, e.g., cholesterol, several SLs, and triglycerides, complicating the pathology of T2D. Thus, targeting multiple metabolic pathways simultaneously to manage glucose, cholesterol, SLs, triglycerides, and body weight may be the answer to control T2D. This has been achieved in a mouse model of T2D: *db*/*db* mice using a biopolymer, encapsulated D-PDMP—an inhibitor of GlcCer and LacCer synthase [79]. Herein, treatment also decreased NPC-1 and mTORC-1 signaling. Previously, mTORC1 has been implicated as a master regulator of cell growth and proliferation [68].

Our current understanding of the role of sphingolipids in diabetes is at the proverbial “tip of the iceberg” stage. Insulin–insulin receptor interaction clearly represents the tip of the iceberg in type II diabetes to regulate glucose levels. However, dyslipidemia involving elevated levels of sphingolipids, e.g., ceramide, LacCer, S1P, GM3 (Figure 4), bulk lipids, e.g., cholesterol, triglycerides, phospholipids and fatty acids, and various proteins involved in cell signaling represents the rest of the iceberg. Thus, it is the sum effects of interactions of these molecules which contribute to diverse phenotypes, e.g., cell proliferation, apoptosis, angiogenesis, inflammation, and oxidative stress, which drive the progression of disease. Among all these phenotypes, inflammation and oxidative stress may play a central and larger role in the progression of diabetes and associated cardiovascular complications and kidney disease than what has been appreciated thus far. Herein, cytokines, oxidized LDL, and various growth factors may raise the level of LacCer, thus collectively contributing to disease progression and pathology.

Metabolomic studies in T2D patients has revealed improvement in the predictive value when sphingolipids were included among other known biomarkers in this disease, and metabolomic studies in T2D diabetic patients have shown LacCer as a valuable biomarker associated with cardiovascular dysfunction, e.g., increased aortic stenosis. This was replicated in a mouse model (ApoE^−/−^) of atherosclerosis. Conversely, lowering the level of LacCer and GlcCer but not ceramide improved AOIMT as well as cardiac function (Figure 4).

In *db*/*db* mice treatment with BPD, a GSL inhibitor, not only reduced the level LacCer but also the blood level of bulk lipids and lipoproteins. Most importantly, treatment reduced blood glucose levels, body weight, inflammation biomarkers, and improved cardiac function.

These studies collectively point to LacCer as an important target to treat T2D and improve cardiac function. Thus, compounds, e.g., BPD, which lower the level of LacCer synthase and LacCer levels in *db*/*db* mice could be tried in diabetic patients with or without metformin to improve cardiovascular and renal health and associated pathologies which accompany diabetes.

## 8. Perspectives

Although experiments in vitro and in mouse models of diabetes suggest a potential role of GM3 in competing with the insulin–insulin receptor interaction and to contribute to T2D, robust evidence using GM3 synthase specific inhibitors is needed.Hyperglycemia and increased levels of LacCer collectively can increase oxidative stress and the generation of advanced glycation end (AGE) products, which is a formidable force in cardiovascular complications and may well adversely affect kidney function in T2D [80].To further understand the complexities in T2D and to develop drugs, a systems-based biology approach is needed that could integrate molecular markers including sphingolipids, genomics, and physiology.

## Figures and Tables

**Figure 1 ijms-23-15442-f001:**
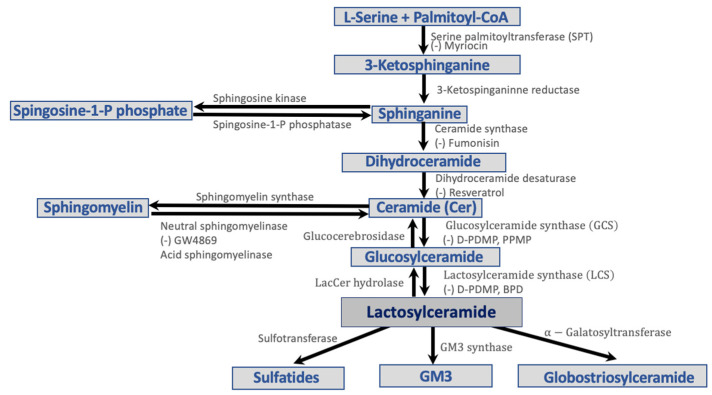
Pathways in the biosynthesis and catabolism of glycosphingolipids and sphingolipid synthesis inhibitors.

**Figure 3 ijms-23-15442-f003:**
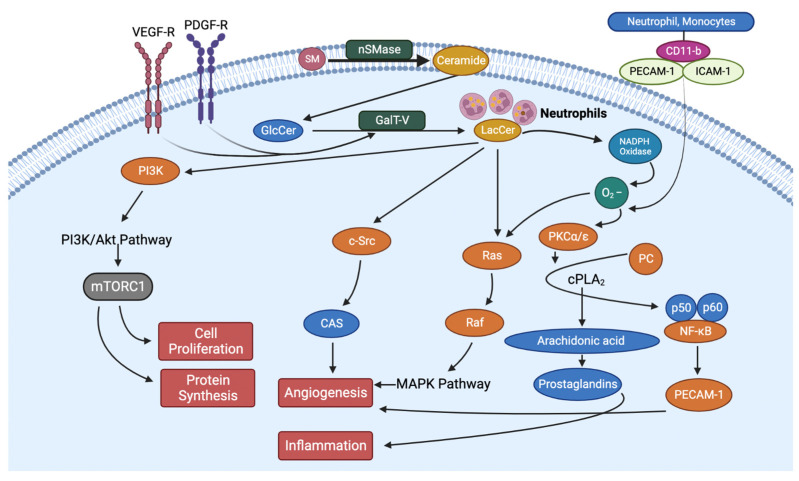
Signaling pathways by which LacCer affects diabetes and inflammation.

**Figure 4 ijms-23-15442-f004:**
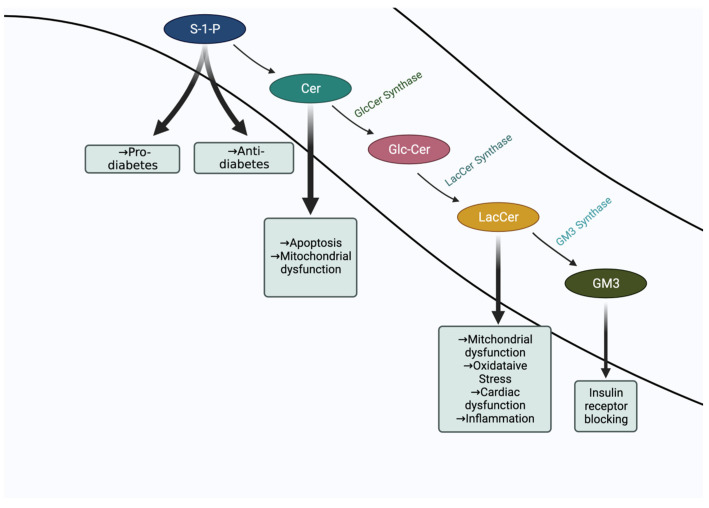
Schema of the Roles of Sphingolipids in Multiple Phenotypes in Diabetes Disease Pathologies.

## Data Availability

Please refer to original articles cited and corresponding journals to have data access. As this is a review article not including data presentation.

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
