# Peer review of "Glycosphingolipids in Diabetes, Oxidative Stress, and Cardiovascular Disease: Prevention in Experimental Animal Models"

_ijms, 2022, doi:10.3390/ijms232315442_

Round 1
Reviewer 1 Report (Previous Reviewer 2)
The authors answered to all my comments including corrections of several references. I think the revised manuscript is worth publishing in International Journal of Molecular Sciences.
Author Response
Please see the attachment.

Reviewer 2 Report (Previous Reviewer 1)
I appreciate the authors' time and efforts to improve the quality and presentation of the manuscript under review from its original version. Expanding the focus and scope of this review paper to include more SLs significantly contributed to a better overall manuscript. However, reading the PDF with the track changes was a little challenging, so I suggest that future revisions submit the final improved version along with the previous version rather than a PDF with track changes.
Author Response
Please see the attachment.

Reviewer 3 Report (New Reviewer)
In this review manuscript, the authors have presented the role of glycosphingolipids on causing T2DM including oxidative stress and cardiovascular diseases. To support the role of glycosphingolipids on causing T2DM, discussed the examples of inhibition using animal model studies. This article is depicting of a high important topic (T2DM) with specific cause of its occurrence. The authors have thoroughly discussed the role of various glycosphingolipids on causing T2DM with rational mechanistic pathways with authentic references. The main question addressed by the author is that one of the reasons for T2DM is due to the increase the level of glycosphingolipids and in this report the have summarized the root cause of it and described the remedies of it showing the literature using animal model studies very authentically.
My comments are as follows:
1. The title of the manuscript is perfectly suited in in line with the manuscript narrative, abstract is found to be very rational and well written.
2. The introduction section is also written very adequately where first part gives us a brief idea about the T2DM and its related complications and in the second part it is well explained about the role of sphingolipids causing T2DM.
3. The schematic representation (Figure 1) about the formation of glycosphingolipids is clearly understandable.
4. The role of GM3 Ganglioside, Sphingosine-1-Phosphate, Glucosylceramides, Sphingomyelin are discussed rationally.
5. Inhibition of LacCer synthesis, /GM3 Ganglioside synthesis to Mitigate the Pathology of Type 2 Diabetes by Reducing Blood Glucose, Body Weight, and Inflammation has been discussed with animal models with authentic references and the results shown in the studies found to be satisfactory.
Query: Is there any T2DM therapeutics by inhibition of glycosphingolipids available till date?
6. Since the authors presented the mechanistic pathways the glycophospholipids like GM3, LacCer etc will cause T2DM related complications, it will be highly helpful to design new therapeutic targets/ drugs for prevention of T2DM.
7. Although present review emphasis the glycosphingolipids in diabetes, it is suggested to mention importance of modern medicine in relation to efforts towards development of antidiabetic medicine. In this regard, it is recommended to emphasis the importance of iminosugars and sugar derivatives as an antidiabetic agents and suggested to cite following relevant articles related to iminosugars in introduction section.
i) Nash, R. J.; Kato, A.; Yu, C-. Y.; Fleet, G. W. J. Iminosugars as therapeutic agents: recent advances and promising trends. Future Med. Chem. 2011, 3, 1513−1521.
ii) Yang, L.-F.; Shimadate, Y.; Kato, A.; Li, Y.-X.; Jia, Y.-M.; Fleet, G.W.J.; Yu, C.-Y. Synthesis and glycosidase inhibition of N-substituted derivatives of DIM. Org. Biomol. Chem. 2020, 18, 999–1011.
iii) Chennaiah, A.; Bhowmick, S.; Vankar, Y. D. Conversion of glycals into vicinal-1,2-diazides and 1,2-(or 2,1)-azidoacetates using hypervalent iodine reagents and Me3SiN3. Application in the synthesis of N-glycopeptides, pseudo-trisaccharides and an iminosugar. RSC Adv. 2017, 7, 41755−41762.
iv) Rajasekaran, P.; Ande, C.; Vankar, Y. D. Synthesis of (5,6 & 6,6)-oxa-oxa annulated sugars as glycosidase inhibitors from 2-formyl galactal using iodocyclization as a key step. ARKIVOC 2022, vi, 5−23.
Overall, after addressing the points mentioned above, I recommend this review to publish in International Journal of Molecular Sciences.
Author Response
Please see the attachment.

This manuscript is a resubmission of an earlier submission. The following is a list of the peer review reports and author responses from that submission.
Round 1
Reviewer 1 Report
Manuscript Title: “Sphingolipid Signaling Networks and Prevention of Diabetes 2
in Experimental Animal Models”
Summary: The manuscript under consideration is a review of the current scientific literature discussing three major concepts regarding specific sphingolipids (SLs), namely GM3, ceramide, lactosylceramide and S1P and their potential involvement in diabetes mellitus pathophysiology. In this regard, the chosen title is somewhat misleading, since the manuscript discuss the role of several different SLs in the development of diabetes type 2, not the respective SL signaling pathways. Moreover, the review includes multiple citations of studies done in different cell types, not only experimental animal models.
This is a well-chosen topic as there are ongoing new discoveries and developments in the field of SLs and their roles in disease processes. Thus, the manuscript will be of particular interest to the readers. The body of the article is well structured and organized in an easy to follow manner. Figures used for visualization of the major concepts are appropriately used. However, since the authors focus on “SL signaling networks”, perhaps including a figure demonstrating these specific networks is needed.
Specific comments and suggestions for improvement:
Introduction section:
- From line 25 to 41 includes poorly written statements about diabetes mellitus with outdated language and classification for this disease. I encourage the authors to use the most recent ADA classification system and re-write this paragraph: “American Diabetes Association. “Classification and diagnosis of diabetes: Standards of Medical Care in Diabetes 2022”. Diabetes Care 2022;45(Suppl. 1).
- Line 52 the sentence starting “It is a state of sensitivity….” Is inaccurate, I believe it should be “insensitivity”. There are other imprecisions in the language used for describing the pathophysiology of DM2. Please refer to the ADA classification system.
- English language and grammar needs improvement through the manuscript.
Section 2: SL Pathway in Diabetes
- The entire section needs serious improvement. The authors begin with “Sphingolipids…” but they actually describe the production of ceramide, which is one of many SL intermediates. I recommend using one of the many previously published reviews written by the doyennes in the sphingolipid research field who pioneered many of the discoveries of the metabolic enzymes, signaling roles of certain lipids etc.
- Figure 1 is incomplete; it schematizes the de novo pathway but not turnover.
- GM3 and it’s metabolism is not included here but appears later in the manuscript without proper introduction.
- Authors introduce the SL metabolic pathway but do not include the signaling pathways in which certain SL act as second messengers. A paragraph introducing the signaling bioactive SLs will improve the quality.
- There are multiple statements that lack proper identification of the tissues or cell types that the effects were observed.
- Line 115: pancreatic beta-cells should express GLUT 2, not GLUT 4, please doublecheck the citation or include explanation.
It is not clear why some SLs such as sphingomyelin (SM) which plays important role in diabetes is not included. SMS1-deficient mice have been shown to have defect in the fusion of the insulin-containing vesicles and release of insulin, for example. Justification needs to be provided for selecting certain lipids but not others.
Reviewer 2 Report
In this manuscript, the authors reviewed the relationship between diabetes mellitus and sphingolipids implicating LacCer as valuable biomarker of type-II diabetes and an important target to treat T2D. I think that the review is intriguing and worth publishing in “International Journal of Molecular Sciences” with minor modifications.
[specific comments]
#1. (page 5, lines 163-165) Reference#39 is a review article and seems inappropriate for the sentence. Refer to the original articles.
#2. (page 5, lines 180 and 188) ”Pro”-diabetic and “anti”-diabetic seem to be used in an opposite way.
#3. (page 9, line 384) Reference#54 is a study using animal models and seems inappropriate for the sentences.
#4. English editing might be recommended throughout the manuscript.
Reviewer 3 Report
In this review, authors describe how different classes of sphingolipids are involved in the development or progression of diabetes, mostly (but not exclusively) in animal models. In general, it includes a good summary of animal studies, using inhibitors of different pathways of sphingolipid metabolism.
The role of some sphingolipids in diabetes is still controversial and requires further research. In this context, the topic could be relevant.
General concept comments
Although the review provides relevant information, it needs to be reorganized.
The text details how different classes of sphingolipids participate in the development or progression of diabetes, mistakenly mixing both concepts. For example, the title refers to the prevention of diabetes, however, most of the processes described in the review point to the progression of diabetes: oxidative stress and cardiovascular risk, not its onset. Both processes are very different, with different factors and mechanisms involved.
For this reason, it is suggested to rearrange the text and highlight the mechanisms that point to the development of diabetes vs. its progression. In addition, it is necessary to include relevant references that are not incorporated, and that could modify the conclusions.
For example: studies about the relationship of GSLs in metabolism and diabetes onset (Ex: Chew et al., JCI Insight 2019 and Berkowitz et al., Front. Cardiovasc. Med. 2022) and intervention studies with sphingolipids in animal models (Ex: Margalit et al, Pharmacol Exp Ther. 2006, Zigmond et al., Am J Physiol Endocrinol Metab. 2009, and Zigmond et al Journal of Inflammation Research. 2014).
Specific comments
- The introduction must be reorganized and limited to only T2D. It should include insulin resistance as a possible cause of hyperglycemia (line 29), and introduce sphingolipids before GSLs (line 60).
- When describing ceramides, it is necessary to mention that they are also in lipoproteins (line 90). The relationship between ceramide and diabetes is correct, but it is disorganized.
- The S1P and diabetes section is unclear, and includes some misconceptions. For example the use of pro-diabetic and anti-diabetic terms seems wrong. The term pro-diabetic should refer to a factor that induces diabetes, but line 180 mentions: Pro-diabetic S1P acts as a mediator to protect against insulin resistance. Please clarify. On the other hand, the pathways of SphK1 and SphK2 are not explained clearly.
- The section about LacCer in diabetes needs to be expanded and discussed. There is a confusion between the development of T2D (due to insulin resistance) and the progression beyond metabolism (oxidative stress and inflammation). None of the cited references demonstrates the role of LacCer in the development of diabetes and metabolic disorders. It is necessary to include references of existing evidence suggesting that simple glycosphingolipids (HexCer and LacCer) could protect against the development of metabolic diseases in cellular, animal and population models (papers mentioned above).
- D-EtDO-P4 should be mentioned in figure 1 (not just BPD).
- Considering that the impact in insulin resistance is clearer for GM3 than for LacCer, it is suggested to paraphrase lines 248-250.
- In line 261 metabolic syndrome is wrongly defined. The text rather refers to atherogenic or diabetic dyslipidemia.
- The association between TG and GSLs according to various studies is inversely proportional, which contradicts line 264.
- By inhibiting the downstream GluCer or LacCer synthesis pathway, it also decreases GM3. Why do you attribute the effects to LacCer? In fact, the description mixes effects of LacCer and GM3. Please clarify section 4.
- In lines 345-350 references are missing.
- In line 384, reference 54 is incorrect.
- The conclusions drawn are not entirely supported by the information in the review, and the title is not representative.